# Practical fluorescence reconstruction microscopy for large samples and low-magnification imaging

**Julienne LaChance**[1], **Daniel J. Cohen**[1,2]*

**1** Department of Mechanical and Aerospace Engineering, Princeton University, Princeton, New Jersey, United States of America, **2** Department of Chemical and Biological Engineering, Princeton University, Princeton, New Jersey, United States of America

* danielcohen@princeton.edu

## Abstract

Fluorescence reconstruction microscopy (FRM) describes a class of techniques where transmitted light images are passed into a convolutional neural network that then outputs predicted epifluorescence images. This approach enables many benefits including reduced phototoxicity, freeing up of fluorescence channels, simplified sample preparation, and the ability to re-process legacy data for new insights. However, FRM can be complex to implement, and current FRM benchmarks are abstractions that are difficult to relate to how valuable or trustworthy a reconstruction is. Here, we relate the conventional benchmarks and demonstrations to practical and familiar cell biology analyses to demonstrate that FRM should be judged in context. We further demonstrate that it performs remarkably well even with lower-magnification microscopy data, as are often collected in screening and high content imaging. Specifically, we present promising results for nuclei, cell-cell junctions, and fine feature reconstruction; provide data-driven experimental design guidelines; and provide researcher-friendly code, complete sample data, and a researcher manual to enable more widespread adoption of FRM.

**Data Availability Statement:** All raw data files are available from the Zenodo database (doi.org/10.5281/zenodo.3783678 All code files are available from GitHub at https://github.com/CohenLabPrinceton/Fluorescence-Reconstruction.

## Author summary

Biological research often requires using fluorescence imaging to detect fluorescently labeled proteins within a cell, but this kind of imaging is inherently toxic and complicates the experimental design and imaging. Advances in machine learning and artificial intelligence can help with these issues by allowing researchers to train neural networks to detect some of these proteins in a transmitted light image without needing fluorescence data. We call this class of technique Fluorescence Reconstruction Microscopy (FRM) and work here to make it more accessible to the end-users in three key regards. First, we extend FRM to challenging low-magnification, low-resolution microscopy as is needed in increasingly popular high content screening. Second, we uniquely relate FRM performance to every-day metrics of value to the end-user, such as cell counts, size, and feature detection rather than to abstract performance metrics from computer vision. Third, we

**Funding:** D.J.C. and J.M.L. received support from 1R35GM133574-01 from the National Institutes of Health (www.nih.gov). The funders had no role in study design, data collection and analysis, decision to publish, or preparation of the manuscript.

provide accessible software tools and characterizations of FRM intended to aid researchers in testing and incorporating FRM into their own research.

This is a *PLOS Computational Biology* Methods paper.

## Introduction

Deep learning holds enormous promise for biological microscopy data, and offers especially exciting opportunities for fluorescent feature reconstruction[1–5]. Here, fluorescence reconstruction microscopy (FRM) takes in a transmitted light image of a biological sample and outputs a series of reconstructed fluorescence images that predict what the sample would look like had it been labeled with a given series of dyes or fluorescently tagged proteins (Fig 1A–1C) [2,6–10]. FRM works by first training a convolutional neural network (e.g. U-Net) to relate a large set of transmitted light data to corresponding real fluorescence images (the ground truth) for given markers[11–13]. The network learns by comparing its fluorescence predictions to the ground truth fluorescence data and iterating until it reaches a cut off. Once trained, FRM can be performed on transmitted light data without requiring any additional fluorescence imaging. This is a powerful capability and allows FRM to: reduce phototoxicity; free up fluorescence channels for more complex markers; and enable re-processing of legacy transmitted light data to extract new information. In all cases, FRM data are directly compatible with any standard fluorescence analysis software or workflows (e.g. ImageJ plug-ins). Such capabilities are extremely useful, and FRM may eventually become a standard tool to augment quantitative biological imaging once practical concerns are addressed.

However, a number of challenges limit FRM accessibility to the larger biological community. Key among these is the difficulty in relating the abstract accuracy metrics used to score FRM to the practical value of FRM data for actual, quotidian biological analyses such as cell counting or morphological characterization. To better appreciate this, consider first that the quality of FRM is typically assessed using a single numerical metric ($P$) such as the Mean-Squared-Error or Pearson's Correlation Coefficient that typically range from (0,1) or (-1,1), and second that it is practically impossible to actually reach perfection ($P = 1$). $P$ can be increased closer to 1 either by training with more images, or by using higher resolution magnification (e.g. 40X-100X) to capture finer details. However, increasing $P$ also carries an intrinsic cost in increased wet-lab and computing time. That improving $P$ is expensive and that $P$ cannot be perfect beg the questions of how good is good enough, and good enough for what (Fig 1D)? For instance, $P = 0.7$ lacks any practical context, and may be quite good enough for a given use case without requiring more work to raise the 'accuracy'. This is why context is extremely important for FRM and why the work we present here focuses on evaluating practical uses of FRM with respect to given $P$ values.

Our goal here is to provide a standardized implementation of FRM and demonstrate its practical performance and limitations for every-day tasks such as nuclear localization and tracking, characterizing cell morphology, cell-cell junction detection and analysis, and re-analyzing legacy data and data collected on different systems (Fig 1). To further emphasize the use of FRM for routine tasks, we will exclusively focus on those lower magnifications (4X-20X) commonly used in high content imaging and cellular screening in contrast to the focus on higher magnifications in prior studies [9,10]. We hope that the included software we developed and the analyses and comparison data we present will help make FRM more

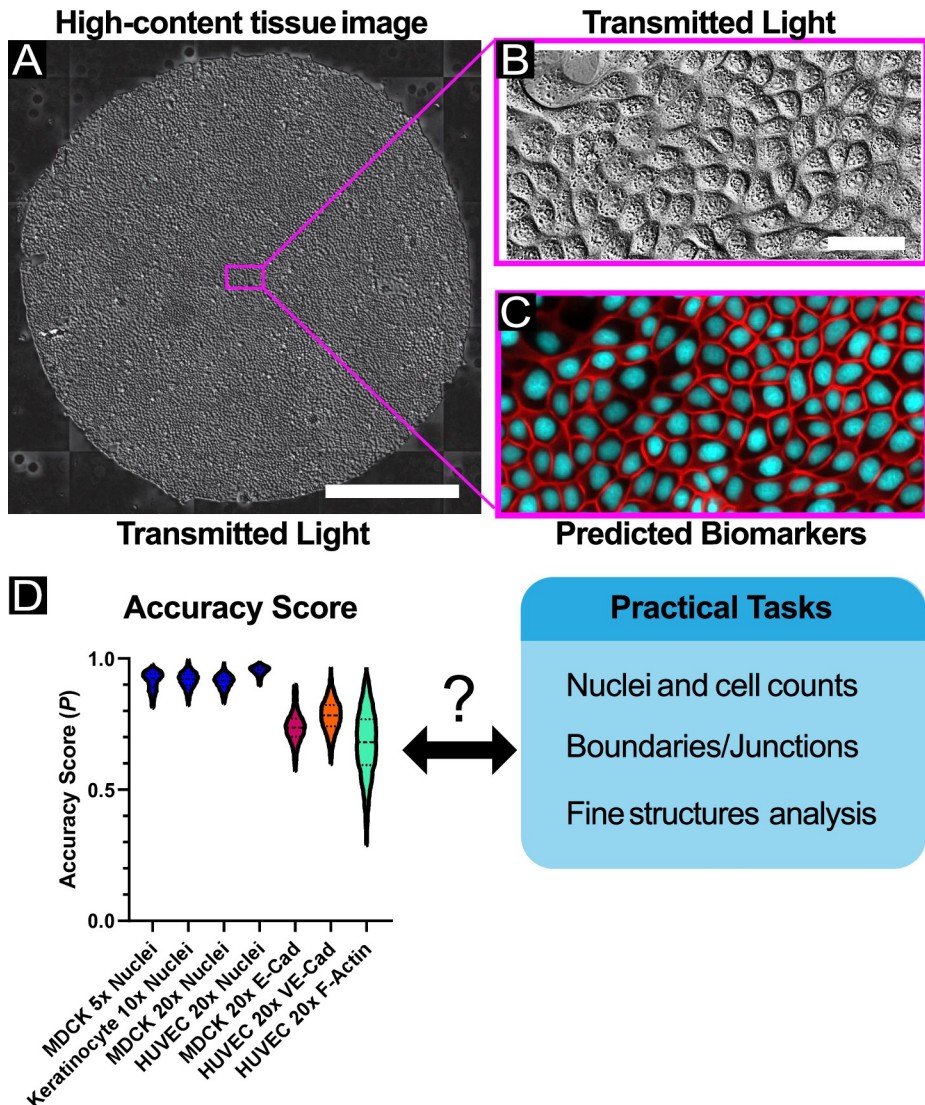

**Fig 1. High-content, high-throughput labeling of fluorescent features.** (A) Sample large tissue of MDCK cells imaged via transmitted light (DIC). The scale bar represents 1 mm. A sub-region of the large tissue is enlarged in B. (B) A representative image which is given as input to the U-Net processing framework. The scale bar represents 50 μm. (C) The predicted fluorescent features (cell-cell junctions and nuclei) produced by the U-Net, corresponding to the same spatial region as in B. (D) Violin plot of accuracy score results from all experimental datasets. $N > 4400$ raw images and $> 20,000$ sub-images for all datasets; see S1 Table for summary statistics. Here, the accuracy score (P) represents the Pearson's correlation coefficient between ground truth images containing positive examples of features and their matched reconstructed images. See *Methods*.

approachable to the broader biological community. To further facilitate this, we have made the entirety of our code and all collected data public as well as providing a full tutorial guide (see Methods and Supplementary Material).

## Adapting U-Nets for Low Magnification, High-Content FRM

While several FRM methods use computationally complex and expensive networks that rely on Z-stacks of images to capture 2D reconstruction[9], other approaches reconstruct 3D image stacks using a modified U-Net architecture[10]. The U-Net itself is commonly used in

machine learning approaches because it is a lightweight convolutional neural network (CNN) which readily captures information at multiple spatial scales within an image, thereby preserving reconstruction accuracy while reducing the required number of training samples and training time. U-Nets, and related deep learning approaches, have found broad application to live-cell imaging tasks such as cell phenotype classification, feature segmentation[10,14–19], and histological stain analysis[20–23].

Our implementation here provides an archetypal U-Net and framework intended for the cell biology community. Briefly, our workflow is as follows. First, we collected multi-channel training images of cultured cells where each image comprised a transmitted light channel and associated fluorescence channels (labeled using genetically encoded reporters or chemical dyes; see Methods). These images were then broken into 256x256 pix$^2$ 'sub-images' in ImageJ and then input into the network. Such image chopping is necessary for the average researcher to account for the average RAM and graphics cards available on standard workstations. Van Valen et al. examine the trade-off between conv-net receptive field size on feature segmentation accuracy versus training time (and therefore over-fitting of the model): optimization of the receptive field size may lead to further performance gains[14]. These data are then passed through the U-Net network to generate trained weights—the pattern recognition side of the network. Here, the transmitted light images serve as input to the network, which is then optimized to minimize the difference between intensity values of the output predicted images and the intensity values from the ground truth corresponding fluorescence images (e.g. Fig 1). This process can be extended to full time-lapse video fluorescence reconstruction, making it well suited for high-content live imaging (see S1–S4 Movies). We have provided all of our code, all raw and processed data, and an extensive researcher manual (DataSpace, GitHub) to encourage exploration of FRM.

As our conventional performance metric, we selected the Pearson's Correlation Coefficient (PCC), which is commonly used in cell biology when comparing the co-localization of two or more proteins, and also used in computer vision to assess spatial-intensity when determining image similarity. However, we observed that naively applying the PCC across our whole dataset skewed the results due to the large number of images containing primarily background (common with high content imaging of oddly shaped or low density samples). This resulted in poor PCC scores as the network tried to reconstruct the pseudo-random background noise. To address this, we report a corrected accuracy score ($P$) representing the PCC of a large subset of images in a given dataset containing positive examples of the feature (nuclei, junctions, etc.) based on an intensity threshold (S1 Fig, see Methods). This approach will improve network performance for datasets containing large amounts of background signal. We also considered a second modified metric, which we call "Segmentation PCC", which reports PCC only on those pixels which exceed an intensity threshold in the ground truth image (S2 Fig, see Methods). However, since many of our key datasets either (1) are used in applications which require background information, such as with nuclei segmentation/tracking, or (2) represent structures on the grayscale spectrum where the background/foreground distinction is highly subjective, such as with MDCK E-cadherin, we found the P metric to be more suited for our analyses. The Segmentation PCC may benefit future researchers in assessing foreground accuracy in alternative FRM applications.

To broadly explore the utility of FRM for high-content imaging applications, we captured transmitted light images using 4X, 10X, and 20X air objectives using either Phase Contrast or Differential Interference Contrast (DIC), and collected data across 3 different cell types—renal epithelial cells (MDCK), primary mouse skin keratinocytes (KC), and human umbilical vein endothelial cells (HUVEC). Variable training set sizes were tested to also explore the effect of data set size on 'accuracy'—a key practical aspect of designing an FRM study. The biomarkers

we trained against comprised a nuclear dye (Hoechst 33342), an F-actin dye (SiR-Actin) and genetically encoded fluorescence reporters for E-cadherin and VE-cadherin. Traditional Accuracy Scores for each of these are summarized in Fig 1D and S1 Table, and we will next present case studies from each of these data sets before concluding with a discussion of how 'accuracy' relates to visual quality to help researchers design experiments for FRM. Specifically, we apply commonly used 'down-stream' analyses from high-content screening to both ground truth and FRM predictions to evaluate how closely analysis of a prediction matches that of the ground truth. While we necessarily show representative data here, we provide the statistical distribution for all accuracy scores, as well as complete 2D cytofluorogram heatmaps of ground truth to prediction mapping across all of our data for each biomarker (S3 Fig).

## Results

### Demonstration of FRM for low-magnification nuclear fluorescence reconstruction and analysis

One of the most common computational image processing needs for screening and low-magnification image is nuclei detection or segmentation, which enables cell counting, time-lapse tracking, and statistical analyses of ensemble distribution and geometry. While a variety of traditional image processing approaches exist to extract nuclei from phase or DIC images, such techniques require extensive fine tuning, ultimately only work for certain cell types, and often fail to work at all with DIC images. The most reliable and standardized technique by far is using a vital dye (e.g. Hoechst 33342 or DRAQ) to stain the nuclei. However, Hoechst requires cytotoxic UV illumination while DRAQ (far-red fluorescence characteristics) has been linked to cell cycle alterations due to its chemistry [24–26]. Both dyes also exhibit loss of signal over extended time-lapse imaging. Alternately, genetic reporters such as H2B nuclear labels can be engineered into cells (e.g. transfection, viral addition, etc.), but this adds more overhead, incurs phototoxicity, and still requires a dedicated fluorescence channel for a relatively simple structure (the nucleus) in lieu of a more complex or useful label. Hence, there is a clear practical benefit to fluorescence reconstruction of cell nuclei, especially in time-lapse imaging where freeing up a channel and reducing phototoxicity are each quite valuable. Further, fluorescent reconstruction of nuclei supports any software or analysis pipeline that might normally be employed with fluorescent nuclei data, meaning that workflows need not be altered to leverage FRM data here.

To validate low-magnification, high-accuracy nuclear FRM, we collected data in both MDCK renal epithelia cells (5X phase contrast, Fig 2A–2D) and primary skin keratinocytes (10X phase contrast, Fig 2E–2H) while using Hoechst to label nuclei and generate our ground truth training data. Representative images are presented as a sequence of phase contrast, nuclear ground truth (green), network predictions (red), and a merged overlay (yellow for a perfect merge).

The Accuracy Score ($P$) is included for context, while the statistical distributions of $P$ for each cell type are presented in Fig 2I demonstrating the actual network performance. The performance with the keratinocyte data is particularly striking given how irregular and poorly resolved the cells appear in phase contrast (confounding traditional segmentation).

The network performs visually well in both cases, with $P \sim 0.9$, but to represent what that means in practice, we quantified disparities in the predictions with respect to nuclear size for geometric accuracy (Fig 2J) and centroid error to reflect positional accuracy (Fig 2K). In both nuclear area cases, the U-Net slightly overpredicts area, likely due to slight noise in the predictions blurring the predicted nuclei and effectively increasing area. However, the distributions from the violin plots are quite similar in structure, and the predictions are well within the

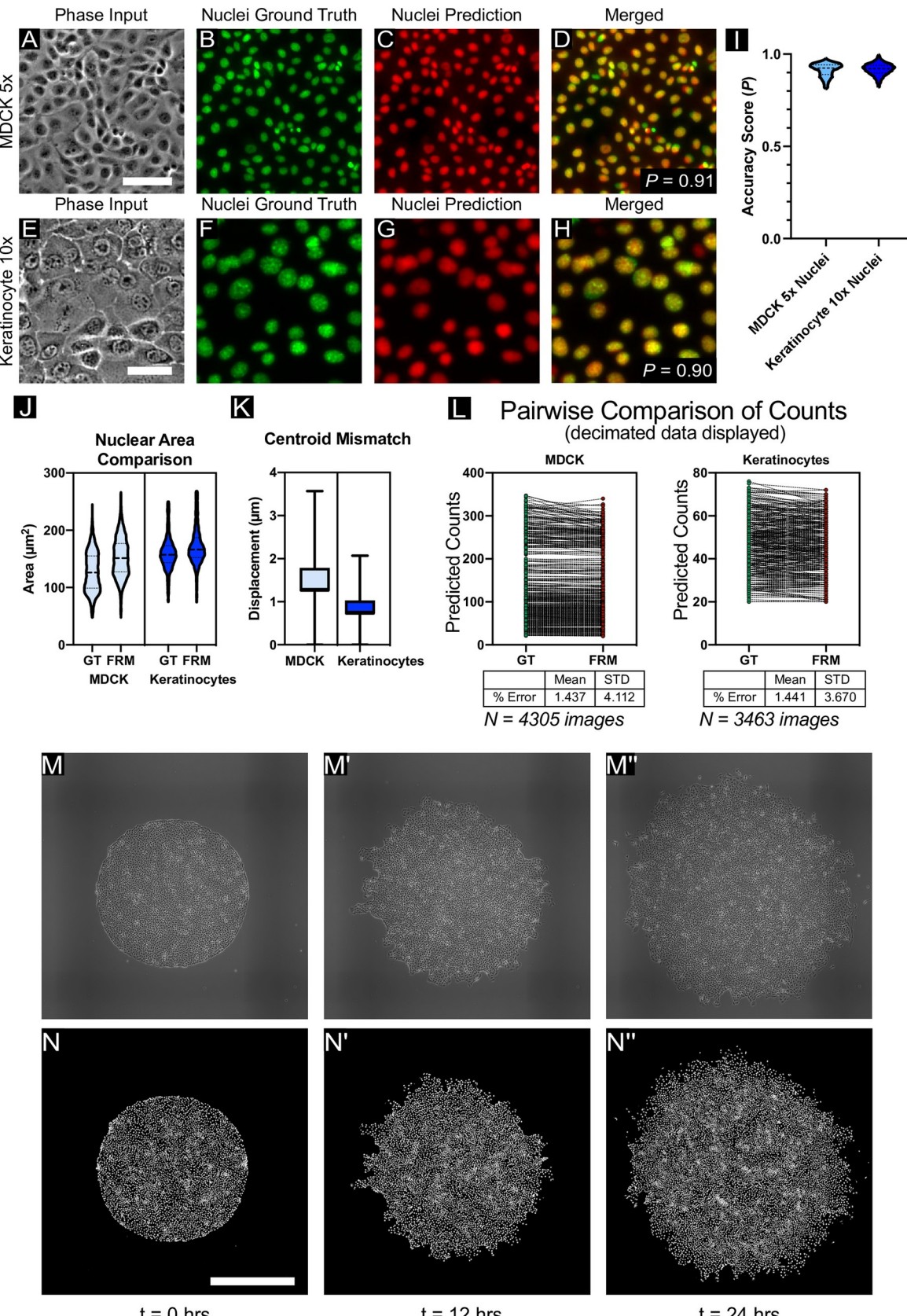

**Fig 2. Low-magnification nuclei reconstruction.** (A) Representative transmitted-light image of MDCK cells at 5x magnification, with corresponding: (B) ground-truth nuclei, stained with Hoecscht 33342 and imaged with blue fluorescent light; (C) nuclear prediction produced by the network; and (D) the overlay of (B) and (C) displayed in red and green, respectively. The raw accuracy score between (B) and (C) is given at right. The scale bar is 100 μm. (E) Representative transmitted-light image of keratinocyte cells at 10x magnification, with corresponding (F, G, H) ground truth nuclei image, predicted nuclei, and overlay, respectively. Scale bar is 50 μm. (I) Comparison of the accuracy score distributions across the 5X MDCK and 10X Keratinocyte datasets, $N > 4400$ test images for each dataset (see S1 Table). (J,K) A comparison of nuclear area estimations and centroid-centroid displacement estimations, respectively, for the two low-magnification datasets considered here. See *Methods*. (L) Pairwise comparisons of predicted cell counts for MDCK (left) and Keratinocytes (right). Summary statistics and $N$ shown below plots. (M-N) Sequence of phase images (M-M") from a time-lapse at 0, 12, and 24 hours of growth, with corresponding nuclear predictions (N-N") respectively. Input data consists of MDCK WT cells imaged at 5x magnification and montaged; the U-Net was applied in a sliding-window fashion to predict small patches of the image in parallel. Scale bar is 1 mm. Higher resolution movies showing the migration dynamics can be seen in S1 Movie.

usable range for practical cell counting and segmentation. With respect to nuclear centroid localization, mean errors span 2 microns (5X MDCK) to 1 micron (10X KCs). The improvement from 5X to 10X can likely be attributed to the resolution increase in the magnification, but in both cases the errors are quite small and more than sufficient for standard cell counting, nuclei tracking, and neighbor distribution analyses. *Finally, we compared basic cell counts using a pair-wise comparison plot (decimated), which demonstrated a mean count error over the entire datasets between ground truth and FRM predictions of ~1.4% (Fig 2L).* Whether a higher $P$ would be beneficial would depend on the specific analysis in question—here, the accuracy is more than sufficient.

As a final demonstration of the utility of low-magnification reconstruction and nuclear tracking, we input legacy data from a 24 hr time-lapse experiment of the growth dynamics of large epithelia (2.5 mm$^2$, 5X) and the network output a reconstructed movie of nuclear dynamics (Fig 2M and 2N, and S1 Movie) compatible with standard nuclear tracking algorithms (e.g. Trackmate in FIJI). Images were captured every 10 minutes, and previous efforts to perform this experiment using fluorescent imaging of Hoechst resulted in large-scale cell death, hence FRM proved highly effective both as an alternative nuclear labeling approach for large-scale, long-term imaging, and as a means to reprocess pre-existing, legacy datasets.

## Reconstructing cell-cell junctions for segmentation and morphology applications

Cell-cell junctions and cellular boundaries in cellular ensembles have implications spanning the epithelial-mesenchyme-transition (EMT), tissue mechanics, and tissue maturation[27–29] and are of broad interest from cellular biophysics to high content screening. However, there are no vital dyes for junctional proteins (e.g. E-cadherin), necessitating either antibodies or genetic reporters. In the absence of a specific marker, cell boundaries are relatively difficult to accurately segment, especially from DIC images (Fig 3A), and proxy techniques such as Voronoi tessellation from nuclei data often fail to capture cell shape and organic features such as curved boundaries. Instead, junctions and boundary data most commonly come from biomarkers such as E-cadherin, so we trained our U-Net using MDCK cells stably expressing E-cadherin:RFP (Ecad:dsRed) and imaging with a 20X/0.75NA objective—a well-balanced objective favored for high-content imaging and immersion-free time-lapse imaging.

The U-Net was able to reconstruct E-cadherin junctions with high visual accuracy, as shown in the sequence from Fig 3A–3D. While $P = 0.74$, the reconstruction is quite spatially accurate, which is unexpected given how difficult it is for humans to detect cell-cell junctions by eye in a DIC image. To better highlight the accuracy and utility of junctional FRM, we explored how the network reconstructed a subtle 3D feature of epithelial junctions where a slanted junction is formed between two cells by one cell pushing slightly under another (the

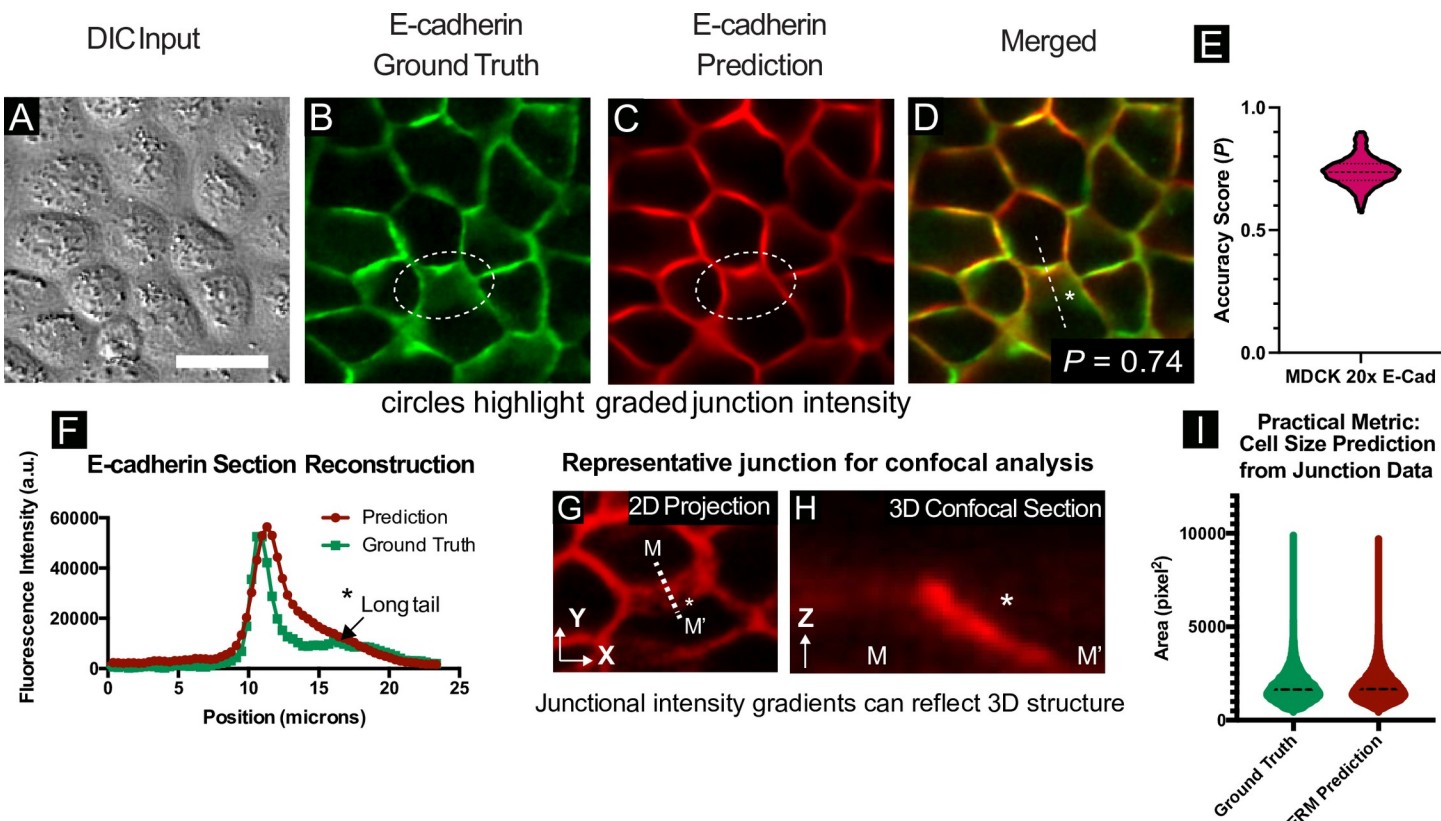

**Fig 3. Cell-cell junction reconstruction from DIC data and capturing otherwise invisible morphology.** (A-D) Images of MDCK WT cells at 20x magnification were processed using a neural network trained to reconstruct cell-cell E-cadherin junctions. Representative ground truth features are shown alongside, and merged with, network predictions. The scale bar is 30 μm. (E) Ensemble statistics for E-cadherin reconstruction; $N$ = 4539 test images, see S1 Table. (F) Line sections from identical spatial regions in (B) and (C) highlight the accuracy of predicted fluorescence intensity across cell-cell junctions (normalized to 16-bit histogram). From 2D transmitted light input (A), 3D structures may be detected. (G, H) Representative cell-cell junction and corresponding confocal section, highlighting the relationship between 2D junction signal and 3D features. '*' in (G) is 7 μm above the basal plane. (I) Practical metric assessing estimated cell areas for entire E-cadherin training set (~30,000 individual cells). Junction segmentation was used to calculate cell areas for ground truth and FRM predictions and the distributions of detected cell areas were plotted for comparison. Ground truth mean area was 1813 pix$^2$; FRM predicted mean area was 1791 pix$^2$.

region enclosed in the dashed oval in Fig 3B–3D). Such slanted junctions may indicate a degree of fluidity or direction of migration and are also impossible to discern by eye. We quantified the accuracy of the FRM image by taking a line section perpendicular to this slanted junction (Fig 3D and 3E) and comparing the profiles of the ground truth and the FRM image. In this line section, intensity values up to and including the peak value are similar, and intensity values exhibit a graded decay within the slanted junction, indicating that the FRM network is able to capture subtle 3D information from the 2D input image. To emphasize the 3D nature of this feature, a representative Z-section from an E-cadherin junction imaged by scanning confocal is shown in Fig 3F and 3G. As a demonstration of a common practical analysis performed using junctional data, we calculated the distribution of cell areas in both ground truth and FRM predictions (see Methods). Shown in Fig 3I, we observed very similar area distributions with mean calculated cell areas over ~30,000 individual cells varying by ~1%. FRM can again be used for high-fidelity reconstruction during a timelapse, allowing both nuclei and junctions to be predicted throughout long acquisitions (see S2 Movie). A large dataset of reconstructed junctions and nuclei can be found in our supplemental data repository (http://doi.org/10.5281/zenodo.3783678, see *Dataset Availability Statement*) for comparison. Overall,

our network captures junctional intensity and geometry, both of which are invisible to the eye in the DIC input image.

## Fine structure reconstruction

In practice, high content imaging is inherently a trade-off between throughput and resolution. The more detail we can extract from lower magnification images, the more efficient the imaging and analysis. Here, we demonstrate the practical performance of FRM and a 20X/0.8NA objective to reconstruct fluorescence signatures for several useful sub-cellular markers using HUVEC cells that stably expressed VE-Cadherin:YFP (mCitrine) and were labeled with Hoechst 33342 (live nuclear dye) and SiR Actin (infrared live actin dye). Processed timelapse data (see S3 Movie) highlights the variation of these fluorescent features given the same input (DIC) image shown in Fig 4A.

As a baseline, we characterized prediction accuracy for cell nuclei as the nucleus itself is relatively low resolution, but detection of sub-nuclear features requires higher accuracy. The Fig 4B column demonstrates FRM performance for 20X nuclei including a line section through both the bulk structure and sub-nuclear granules. Visually, the FRM image is quite accurate, and $P = 0.91$ in this case. The line section easily captures the bulk form of the nucleus, but does not quite capture the texture inside the nucleus, although it does capture the rough form.

Next, we trained the network on identifying Actin after first staining HUVECs using the SiR-Actin live imaging dye. Here, the column in Fig 4C shows significantly reduced performance as the fine F-actin filaments visible in the ground truth fail to be reconstructed in the predictions ($P = 0.67$) with the exception of some of the cortical filaments at the very edge of the cells (see the line profile). We hypothesize this is primarily due to fundamental limitations of DIC imaging and the lack of contrast for intracellular F-actin, but it may also be due to the network overprioritizing cortical filaments and the diffuse cytoplasmic signal. However, in practice we found that these FRM data were useful for general cell body detection and potential segmentation analyses due to the relatively homogeneous reconstructed fluorescence in the cytoplasmic space.

Finally, we trained the network with VE-Cadherin:YFP data in an attempt to reconstruct not only cellular borders, but also the well-characterized, nano-scale membrane fingers that develop in endothelial cell-cell junctions and indicate the direction of front-rear polarity in each cell [30]. In contrast to the actin performance, FRM proved far more capable here and readily detected both general VE-Cadherin boundaries (Fig 4D column) and the membrane fingers (Fig 4E column), although $P = 0.77$ still seems quite low and likely relates to the network attempting to reconstruct the more variable granules in the center of the cell, which are irrelevant for junctional analyses. While VE-Cadherin protrusions and boundaries are sometimes detectable by eye in DIC (as in Fig 4A), they are still quite subtle in the best case, and developing a traditional computer vision process to detect and extract them has not been demonstrated, thereby again highlighting the practical value of FRM to reconstruct not only fluorescence, but also key morphological markers that are much easier to analyze in the FRM image than in the DIC image. The statistical accuracy distributions are shown in Fig 4F, where the spread of the data in F-actin indicates the lack of reliability, while the tighter distributions for nuclei and VE-cadherin indicate more useful reconstructions. Again, the value of FRM depends on the specific question and context, and the decision of whether it is 'good enough' at detecting fine structures rests with the end researcher. For reference, short movies of several markers are presented in S3, S4, and S5 Movies, and a large set of representative images can be found in our supplemental data repository. As a final note, training the network with large and varied datasets enabled it to begin to predict statistically rarer events, such as mitotic divisions,

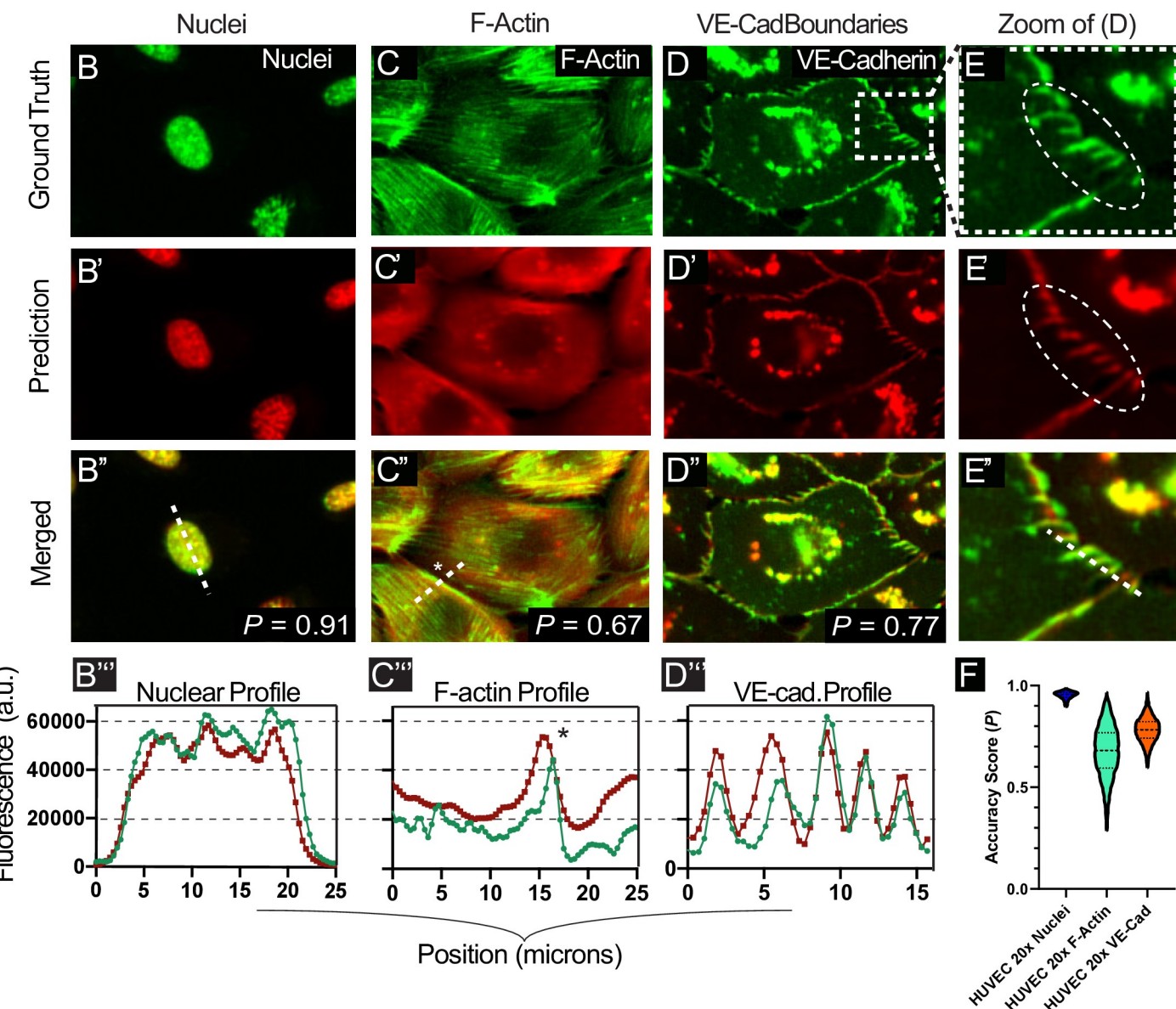

**Fig 4. Coarse-to-fine feature reconstruction.** (A) A representative transmitted-light image of HUVEC cells at 20x magnification with its corresponding structures of varying scale (B-D). The scale bar represents 30 μm. (B-D) the relatively large nuclei, the finer VE-Cadherin structures, and thin F-Actin filaments which are not readily resolved by the network. Ground truth fluorescent features are displayed alongside, and merged with, network predictions. (E) displays zoomed-in portions of images shown in (D). Line sections from (B", C", and D") are displayed graphically in (B''', C''', D'''), to enable intensity comparisons across the ground truth and predicted features. (F) summarizes the distribution statistics, clearly showing the uncertainty in F-actin, with tighter reconstruction for nuclei and VE-cadherin. N = 5500+ test images for these datasets.

as shown in S6 Movie, indicating the benefits of training against time-lapse data where dynamic phenomenon have a higher chance of occurring over time.

## Comparing FRM visual performance to P scores, training set size, and convergence

A key feature of FRM is that its performance can often be increased by collecting more training data, which in turn ought to improve $P$. However, $P$ will never be perfect, nor is $P$ necessarily the best metric to go by when determining if an FRM image is 'good enough', as clearly the context matters and the key question is 'good enough for what?' Hence, we sought to provide several examples of how the size of the training set affects both $P$ and the actual visual accuracy or quality of the resulting FRM predictions.

To do this, we first swept through different sizes of training sets for many of the biomarkers presented earlier (see *Methods*). Briefly, we selected random subsets of very large datasets and trained the U-Net from scratch with these subsets. This process was repeated for different fractions of the complete dataset to capture the FRM performance versus training set size. The relationship between $P$ and training set size is shown in Fig 5A and 5B, where we show both how a single camera images is first chopped into 16 (typically) sub-images for analysis (Fig 5A), and how $P$ and training set size vary relative to either camera images or sub-images (Fig 5B). These data demonstrate that the rate of change in quantitative quality ($P$) vs. training set size is monotonic across different biomarkers, with more data yielding higher accuracy scores with diminishing returns. To relate these data to a more practical readout, we demonstrate how nuclear count performance depends on training set size using our 5X and 20X MDCK datasets (Fig 5C and 5D). These data emphasize both that FRM performance variance goes down with larger training sets, but also that relatively small numbers of training images can produce useful results.

For a clearer visual comparison of performance, we also provide FRM results from the reduced training sets for the 20X HUVEC and MDCK data (Fig 5E and 5F; respectively). Here, the input and ground truth data are presented alongside representative FRM predictions from networks trained with different numbers of training images (noted below each image). As a single image from the camera is split into sub-images for training, a fractional image (e.g. Fig 5E, 1/16$^{th}$ column) implies that the network was trained on just a small crop from a single micrograph. Again, these data all indicate that FRM quality varies directly with the size of the training set, as expected.

Further, the actual predicted images shown in Fig 5E and 5F offer further nuance because they demonstrate that training the network against even a single image could be sufficient to capture nuclei for the purposes of tracking or segmentation, while just 6 images would be sufficient to capture cell shape and junctional geometry in epithelia assuming the researcher were willing to perform some simple manipulations such as background subtraction. There is an obvious performance increase for both cadherins when the training set comprises several hundred images, but it is difficult to visually detect a difference between nuclei reconstructed from 6 or 400 images.

We next assessed how the number of training epochs (and associated convergence) affected network performance. This is valuable because some researchers may have a great deal of training data but limited computational resources. Here, we present data from several key datasets in Fig 6 where identical U-Net models were trained for a fixed number of epochs (1, 5, 10, 50, 100, or 500 epochs) and the accuracy score ($P$) (Fig 6A). Additionally, we report the actual training hours (Fig 6B) based on our specific hardware (NVIDIA Tesla P100 GPU). Again, we relate these data to practical performance using nuclei counts, this time plotting

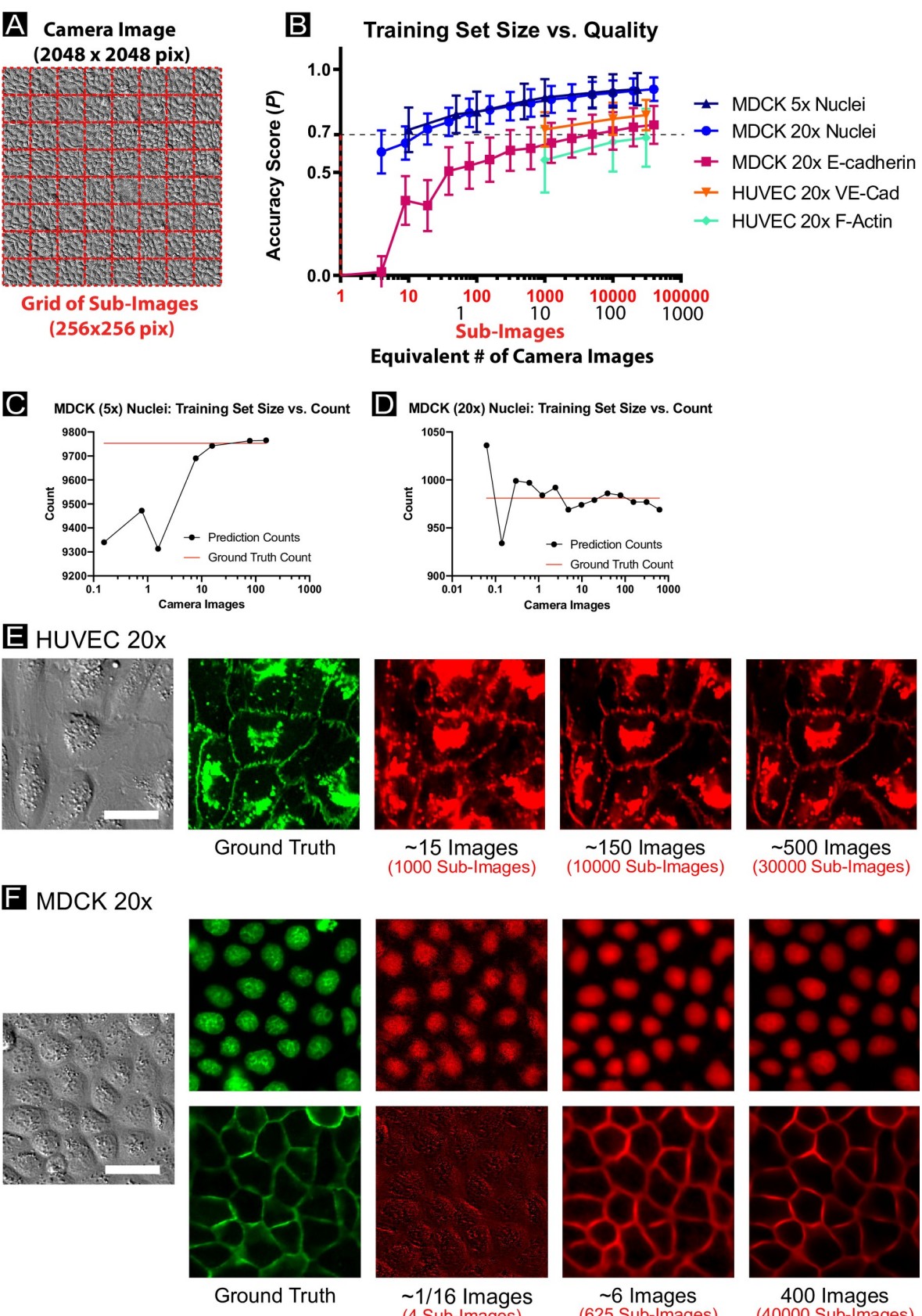

**Fig 5. Impacts on prediction accuracy from smaller training sets.** (A) Cropping a sample image into 64 sub-images. (B) A comparison of network prediction accuracy as a function of training set size. The U-Net is trained with the complete dataset as described in S1 Table for each experimental condition. Then, random images representing a fraction of the total training set is used to train a new U-Net from scratch. The average number of cells per sub-image were: MDCK 5x: 107 cells per sub-image; Keratinocyte 10x: 26 cells per sub-image; MDCK 20x: 11 cells per sub-image; HUVEC 20x: 3 cells per sub-image. (C, D) display nuclei counts for representative images of MDCK cells (5x, 20x magnification, respectively) as a function of training set size, with each model independently trained. Practical readouts such as nuclei count may vary widely with small training set size. (E, F) display representative images for the HUVEC 20x dataset and the MDCK 20x dataset, respectively, with predictions shown for various training set sizes. All scale bars represent 30 μm.

how the distribution of counts evolve as a function of training epochs and relative to the ground truth distribution (Fig 6C and 6D). Overall, increasing training epochs correlates with improved performance, as highlighted visually in Fig 6E and 6F where the evolution of predictions of nuclei and E-cadherin are presented relative to training epochs. In general, researchers limited by computational bottlenecks may opt to reduce training time as the network converges to 'good enough' results as a function of their practical measure of performance. To emphasize this, consider Fig 6E, where the dashed circle indicates how regions of higher cell density reconstruct poorly for low training epochs; highlighting the importance of assessing practical model performance on a range of potential biological and imaging conditions.

As a final analysis, we considered improving FRM performance by altering the network architecture. Here, we first compared the standard U-Net to a neural network architecture which was essentially two U-Nets stacked end-to-end with additional residual connections. Such an approach has been shown to improve network depth and performance in other applications[31–33]. Here, however, we observed no benefit to training a deeper network (see S4 Fig). Further, given the significant temporal and computational cost, we advise against its use for this kind of FRM. While no substantial benefit was observed as the result of data augmentation (flip, rotation, zoom) on standard U-Nets trained with large and varied datasets (see S5 Fig), researchers should test data augmentation techniques on their own data to assess accuracy gains prior to widespread application. Alternately, we explored the role of the loss function, testing our Pearson's-based loss function against the traditional Mean-Squared-Error loss function and found no significant difference (S6 Fig; Methods). Hence, we conclude that our minimal U-Net implementation performs well as a foundation for a variety of daily analysis tasks without requiring significant fine tuning.

## Discussion

### Limitations of existing accuracy metrics and the importance of context

Our data—collected from actual, real-world analyses, highlight the limitations of using traditional accuracy metrics from computer vision for biological image analysis. Specifically, while there is a general relation between an improved $P$ Accuracy Score and FRM quality, it is not linear nor intuitive how to determine what is 'good enough' given only a $P$ value devoid of context for a specific analysis. Further, and most critically, FRM does not reconstruct images according to human imperatives. The U-Net only optimizes via the specific loss function it has been given (e.g. Mean-Squared-Error or the Pearson's coefficient). What the computer considers 'good' need not match our own assessments of value and quality.

As a practical example, compare the FRM performance for E-cadherin ($P = 0.73$; Fig 3) and F-actin ($P = 0.67$; Fig 5). While the accuracy metrics differ by $< 10\%$, the FRM of F-actin only detected peripheral actin cables, otherwise blurring all internal features into a homogeneous signal. Nonetheless, even this plainly 'inaccurate' signal could prove useful for cytoplasmic reconstruction and tracking. In stark contrast, the E-cadherin data was much more visually accurate and also captured key quantitative features of the ground truth such as junctional

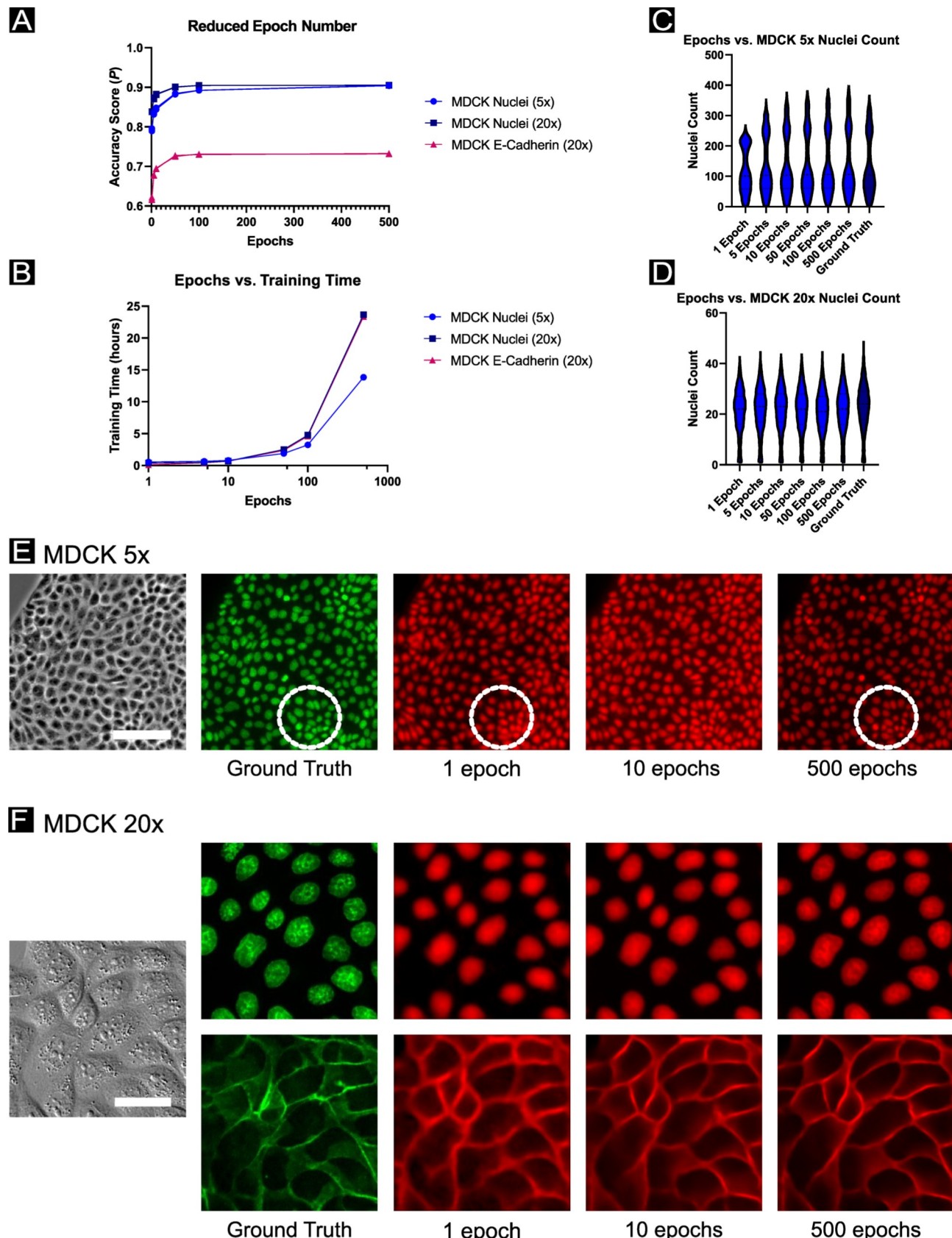

**Fig 6. Impacts on prediction accuracy from reduced number of training epochs.** (A) A comparison of network prediction accuracy $P$ as a function of the number of training epochs. A standard U-Net model is re-trained from scratch for each experimental condition (1, 5, 10, 50, 100, 500 epochs), with $n = 3$ per condition (with accuracy scores overlaid in the plot and visually similar). (B) The time in hours to train the reduced epochs models shown in (A), on an NVIDIA GeForce GTX 1070 Ti GPU. (C, D) display nuclei counts for the segmented prediction images compared to the segmented ground truth image, with predictions produced from the reduced epoch models corresponding to the MDCK datasets at 5x and 20x magnification, respectively. (E, F) display representative images for the MDCK datasets at 5x and 20x magnification, respectively, with predictions shown for various numbers of training epochs. The dotted circle indicates a region of higher cell density; in such regions, practical readouts such as nuclei count from segmentation may vary widely with lower training epoch number. Scale bar in (E) represents 100 μm; scale bar in (F) represents 30 μm.

localization and intensity, and even the subtle intensity gradients representing 3D morphology despite having only a slight improvement in $P$-values. However, a score of 0.73 is far enough from '1' that it is ambiguous in the absence of a specific analysis, which is why FRM must be evaluated in the context of a given question or analysis, since it will never be perfect.

Our work also highlights how FRM and similar machine learning techniques need performance benchmarks and metrics that are both standardized and, critically, context-aware. For instance, although FRM has a relatively low annotation burden (as the fluorescent channel essentially auto-annotates the transmitted light channel) for training purposes, there still remains the problem of how the researcher knows how a given FRM prediction performs against benchmarks of practical value to that researcher. Critically, we demonstrated here that common, but abstract, benchmarks often fail to clearly reflect the practical value of that FRM reconstruction for the average biologist. In other words, what is 'mathematically good' (e.g. high P, etc.) does not adequately reflect what is 'practically useful'. Hence, while FRM is clearly powerful, it is like any other tool and requires that the researcher deploy and assess it intelligently. Establishing clearer benchmarks, challenges, and more approachable performance metrics would be of great value towards transitioning FRM and similar techniques into standard practice.

## Practical considerations for training on new, low-magnification data

We specifically targeted the lower-magnification end of the imaging spectrum to explore how well FRM performed at magnifications more commonly used for high content imaging applications such as timelapse studies of very large cellular colonies or massive screens using multi-well plates. Our data indicate that such magnifications can be effectively combined with FRM for applications spanning nuclear tracking, cell-cell junction analysis, and certain fine-structure reconstruction even at just 20X.

A particular concern for the average researcher of a complex machine learning process is the size of the dataset required as this can impose potentially strenuous experimental demands. However, our characterization of FRM performance vs. data set size again shows the importance of context as relatively few images are needed to get quite accurate nuclei reconstruction, while a greater number of images are needed for junction reconstruction (Fig 5). However, we also note that our largest training set size comprised at most 500 camera images at 20X (approximately one six-well plate)—something easily obtained with a standard automated microscope, and still compatible with manual capture. Further, a very common approach in machine learning is to 'augment' an image dataset by performing reflections and rotations on images such that the network perceives each augmented image as a different datapoint, thereby virtually increasing the size of the dataset. We did not perform such augmentation here for the sake of simplicity and transparency, which suggests that significantly smaller datasets, if augmented, could still produce good results.

As a final note, it would be valuable to be able to transfer a trained network, either from one optical system or laboratory group with collaborators who might have a different system, or across different cell types. We evaluated the former by applying a network trained on a Zeiss

microscope (5X phase objective, 6.5 μm/pixel camera) to data captured on a Nikon microscope (4X phase objective, 7.3 μm/pixel camera). While we found that applying a scaling factor to make the effective microns/pixel of the Nikon data match the Zeiss data was able to achieve adequate nuclei reconstruction for basic counting, we by far recommend using networks specifically trained on a given optical system in order to improve reliability (see S7 Fig). We also considered the application of trained networks across different cell types; comparable cross-cell-line analyses have previously been performed[34]. S8 Fig presents such cross-network results indicating sufficient performance for nuclei predictions that researchers unfamiliar with FRM could begin by testing pre-trained networks on their own data. As with cross-platform applications, we recommend researchers train a model using data which best captures the conditions under which they expect the model to be applied- including consistent cell types.

### FRM versus machine-learning segmentation approaches

FRM is a quickly developing technology that exists alongside another popular approach where machine learning is used for feature segmentation[14,19,35]. In the latter case, the network is trained to specifically detect 'features' (e.g. nuclei) as binary objects, whereas FRM instead reconstructs the effective fluorescent image of what a fluorescent label against that structure might show. Both are useful techniques, and the best approach depends on the application. However, there are several unique advantages to FRM. First, reconstructing an effective fluorescent image from auto-annotated data (e.g. chemical dyes, antibodies, fluorescent proteins) obviates the need for any manual annotation or pre-processing—often quite time consuming and subjective. This means that an FRM image can be directly incorporated into any existing analysis pipeline intended for fluorescent images, including traditional threshold-based segmentation approaches. Further, more of the original data is preserved in an FRM image, allowing the capture of things such as fluorescence intensity gradients (e.g. Fig 3), and features that might be lost during traditional binary segmentation.

### Concluding remarks

Here, we characterize the value of fluorescence reconstruction microscopy (FRM) for everyday analysis tasks facing researchers working with cell biology. We specifically highlight the need for individual researchers to explore and evaluate FRM in the context of specific research questions rather than accuracy metrics. We also highlight the surprisingly good performance of FRM even with lower magnification imaging or relatively fine structures such as VE-cadherin fingers. Finally, we have made all of our tools and all training datasets publicly available to improve accessibility and provide a starting point for researchers new to FRM to easily explore it for themselves and to eventually build on and improve.

## Methods

### Ethics statement

Our study involved standard mammalian cell type the use of which is approved via Princeton IBC committee, Registration #1125–18. MDCK-II cells stably expressing E-cadherin were received from the Nelson laboratory at Stanford University [36], while the wild-type can be obtained through ECACC. HUVEC cells stably expressing VE-cadherin were received from the Hayer laboratory at McGill University [30], originally based on HUVECs provided by Lonza. Primary murine keratinocytes were isolated by the Devenport Laboratory under a protocol approved by the Princeton University IACUC and shared with the Cohen laboratory directly.

## Tissue culture

MDCK-II (G-type) cells stably expressing E-cadherin:dsRed were cultured in low glucose DMEM. The MDCK-II culture media was supplemented with 10% Fetal Bovine Serum (Atlanta Biological) and penicillin/streptomycin. HUVEC endothelial cells stably expressing VE-cadherin:mCitrine were cultured using the Lonza endothelial bullet kit with EGM2 media according to the kit instructions. Primary murine keratinocytes were isolated from neonatal mice (courtesy of the Devenport Laboratory, Princeton University) and cultured in custom media[36]. All cell types in culture were maintained at 37˚C and 5% $CO_2$ in air.

## Preparation of training samples

We collected training data using 3.5-cm glass-bottomed dishes coated with an appropriate ECM. To coat with ECM, we incubated dishes with 50 μg/mL in PBS of either collagen-IV (MDCK) or fibronectin (HUVEC, primary keratinocytes) for 30 min 37˚C before washing 3 times with DI water and air drying the dishes.

In order to contain a variety of conditions within a single plate to ensure a broad training sample, we placed silicone microwells into the dishes as described in [37] at densities from [1-$2x10^6$ cells/mL] which ultimately allowed for single cells, low density confluent monolayers, and high density confluent monolayers to be captured. Silicone microwells consisted of 3x3 arrays of 9 mm$^2$ microwells into which we added 4 μL of suspended cells in media, allowed them to adhere for 30 min in the incubator (6 hrs for keratinocytes), added media and returned them to the incubator overnight prior to imaging. To further ensure variability, several dishes were also randomly seeded with cells for each cell type.

## Fluorescent labeling for ground truth data

We used the live nuclear dye NucBlue (ThermoFisher; a Hoechst 33342 derivative) with a 1 hr incubation for all nuclear labeling. We used SiR-Actin (Spirochrome) at 10 μM for live F-actin labeling in HUVECs. All other labels were genetically encoded reporters as described.

## Image acquisition

5X MDCK data was collected on a Zeiss (Observer Z1) inverted fluorescence microscope using a 5X/0.16 phase-contrast objective, an sCMOS camera (Photometrics Prime) and controlled using Slidebook (Intelligent Imaging Innovations, 3i). An automated XY stage, a DAPI filter set, and a metal halide lamp (xCite 120, EXFO) allowed for multipoint phase contrast and fluorescent imaging. All epifluorescence imaging was performed using a Nikon Ti2 automated microscope equipped with a 10X/0.3 phase objective, a 20X/0.75 DIC objective, and a Qi2 sCMOS camera (Nikon Instruments, 14-bit). This same system was also used for the cross-system comparison in S7 Fig. Note that images captured on the Nikon were natively 14-bit, so they were normalized to the 16-bit histogram in ImageJ to make them compatible with standard intensity analyses as would be used by many biologists.

Time-lapse imaging effectively increased dataset size as long as sufficient time was allowed between frames to avoid overfitting in the U-Net. MDCK data was collected at 20 min/frame, while HUVEC and keratinocytes were given 60 min/frame. Standard DAPI, CY5, and YFP filters sets were used. Confocal sections of E-cadherin fluorescence in MDCK cells (Fig 3) were collected using a Leica SP8 scanning confocal tuned for dsRed excitation/emission.

All imaging was performed at 37˚C with 5% CO2 and humidity control. Exposures varied, but were tuned to balance histogram performance with phototoxic risk. Data with any visible sign of phototoxicity (blebbing, apoptosis, abnormal dynamics) were excluded entirely from training.

## Data pre-processing and training

Prior to input to the network, raw images were segmented into 256x256 pixel$^2$ sub-images, ensuring consistent slicing across the transmitted-light image and the corresponding fluorescent image. The splitting times are reported as a function of image size (see S9 Fig); this plot was generated by cropping the same image into smaller TIFF images, and applying the same splitting procedure to chop the image into 256x256 pixel$^2$ sub-images. Input images were then normalized by statistics collected across all images in each channel: that is, by subtracting from each image the mean and dividing by the standard deviation. A test-train split was applied, such that a random 20% of the total images were held out to comprise the test set. Additionally, 10% of the training data subset were held out for validation as is standard.

The U-Nets shown in Fig 1 were implemented and trained in Python using the Keras/Tensorflow [38,39] packages, since we find these to be more researcher-friendly for high-level deep learning implementations than alternative open-source deep learning libraries (e.g. PyTorch). Training was performed using the Tensorflow ADADELTA optimizer[40]. In the standard training experiments, the mean squared error (MSE) loss function was applied across pixel intensity values in the predicted images compared to intensity values in the ground truth images. Results from the MSE were contrasted with results from two networks trained to maximize the Pearson's correlation coefficient (PCC). The PCC is commonly used in cell biology for evaluating the colocalization of two fluorescently labeled structures. The PCC loss function was defined, for two intensity datasets $R$ and $G$ as:

$$PCC = \frac{\sum_i (R_i - \bar{R}) \times (G_i - \bar{G})}{\sqrt{\sum_i (R_i - \bar{R})^2 \times \sum_i (G_i - \bar{G})^2}}$$

$$Loss = \frac{(1 - PCC)}{2}$$

Sample training loss plots are provided (S10 Fig), reflecting the use of early stopping during training. That is, when the validation loss did not decrease for 75 epochs, the training process terminated. The training and test set sizes and results are provided for all experimental conditions in S1 Table, and timing data as a function of epochs is given for key datasets (see Fig 6).

For several of the experimental conditions, networks were trained using subsets of the original dataset. To do so, the network architecture was fixed, and networks were trained from scratch by using a random subset of matched input-output image pairs from the original training set. The training setup, including hyperparameters, was unmodified regardless of training set size. The original test set was used to compare results for all training set size tests relative to each experimental condition. Likewise, for several experimental conditions, data augmentation was applied in the form of flipping (both horizontally and vertically), rotation (in increments of 45 degrees), and zoom (up to 20% scale). For flips and zoom, built-in Tensorflow functions were utilized. However, the raw data was re-processed to apply random rotations in order to avoid boundary problems using ImageJ/FIJI, resulting in a modified training set which was five times the size of the original dataset.

## Data testing and image processing

The Pearson's correlation coefficient (PCC) of each test set is determined by individually computing the PCC between each predicted image, as output by the network, and its corresponding ground truth image. Additionally, an accuracy score (*P*) based on the PCC was devised to

more reliably represent the performance of the network. To determine P, we report the PCC on a subset of the test set which selects for only those test images containing positive examples of features (nuclei, junctions, etc.). We construct this subset by manually determining threshold values to distinguish image intensities indicating the presence of the relevant features versus background noise for each test set. That is, the histograms of a subset of the data containing positive examples of features are plotted, and an approximate lower bound on intensity values is estimated to distinguish the features from the background. Then, the histograms of a subset of the data containing only background are plotted to ensure that the threshold value is adequate to label the images as background-only images. The MatLab function rmoutliers() was utilized to remove outliers when $P$ is reported for each condition unless otherwise stated. The segmentation PCC score is determined by using the same intensity threshold used in the $P$ analysis, and determining which pixels in each ground truth image exceed that intensity value in order to produce a mask image. Then the standard PCC is reported against the pixels in the ground truth image which correspond to that mask and the pixels in the prediction image which correspond to the mask, and this process is repeated across the entire test set.

For certain experimental conditions, a nuclear count or area comparison was performed between corresponding ground-truth and predicted images. Initially, both pairs of output nuclear images were segmented independently using standard Gaussian blur, auto-thresholding, watershedding, and size exclusion (to exclude clusters) in ImageJ/FIJI, and then outliers were removed using the MatLab function rmoutliers() when nuclear area was reported. We additionally report the centroid-centroid displacement values for the same low-magnification segmented images. The ImageJ/FIJI plugin TrackMate was used to determine displacements between the ground truth and predicted images, as if they were two frames of a video. Standard TrackMate settings were used and outliers were removed using the MatLab function rmoutliers() for reporting.

When intensity plots for line slices are reported, a line is selected as an ROI in ImageJ/FIJI, and intensity values are exported for analysis. Cell areas were calculated using either fluorescence E-cadherin junction data or FRM predictions of the same using a large image containing ~30,000 individual cells. We opted to perform analysis in entirely in FIJI to better represent what the average researcher might use. We performed the following steps to calculate and compare cell areas. Image pairs were first histogram-normalized to 16-bits. Images were then segmented manually using standard intensity thresholding in FIJI with the same threshold (~30% cut-off). Binary skeletonization was used on both ground truth and FRM images to produce cellular outlines. Outlines were then closed using a single dilation operation. After inverting the images, a standard 'Analyze Particles' in FIJI was used to calculate areas inside the boundaries. These distributions were then plotted to allow comparison.

New large transmitted-light images were processed using a sliding-window technique. We processed a large image by analyzing 256x256 pixel$^2$ patches of the input image with a stride of 64 pixels in each direction. Additionally, the border of each predicted patch was excluded in the sliding-window process, as features near the patch borders are likely to have lower accuracy (often as a function of cells being cut off). The sliding-window predictions at each pixel were then averaged to produce the final large predicted image. Timelapse movies can be processed on a frame-by-frame basis. If scaling was required as described in Fig 3, the input was scaled in FIJI and then passed to the network for analysis. On a standard desktop with an NVIDIA GeForce GTX 1070 Ti GPU, the time to read in, process, and save out a single 256x256 pixel$^2$ image was 0.49 seconds, and the time to read in, split, process, stitch, and save out a 1024x1024 pixel$^2$ image was 10.3 seconds.

## Supporting information

**S1 Fig. A comparison of experimental results in Pearson's Correlation Coefficient (PCC), versus a modified accuracy score *P* (see Methods).** To ensure a fairer comparison, outliers were not removed; only intensity thresholding was performed to produce the modified *P* from the PCC. By filtering the PCC results by an intensity threshold in the fluorescent images, we remove low-scoring background images, which bias our accuracy score on the complete dataset. Visual inspection of the plot reveals the low-scoring images as "bumps" near 0.0. S1 Table summarizes the statistics.
(PDF)

**S2 Fig. A comparison of the segmentation PCC metric against prior accuracy scores (see Methods) on key datasets.** (A, B) The PCC score for the representative "Ground Truth" and "Prediction" images shown in (C, D), respectively, assessed on those ground truth pixels which exceed an intensity threshold value, as the intensity threshold varies. (A) details the segmentation PCC vs. intensity threshold for an MDCK nuclei example, while (B) details this for an MDCK E-cadherin example, both at 20x magnification. The E-cadherin results do not vary smoothly with intensity threshold. (C, D) Matching ground truth and prediction images for MDCK 20x nuclei and E-cadherin examples, respectively. Segmentation masks are shown corresponding to three different intensity threshold levels. According to the choice of intensity threshold, pixels representing sub-cellular features may be excluded or undesirable background pixels may be included. (E) Three accuracy scores (PCC, *P*, and segmentation PCC) were compared across three datasets (MDCK nuclei at 5x and 20x magnification, and MDCK E-cadherin at 20x magnification). Outliers were not removed.
(PDF)

**S3 Fig. Intensity-intensity heatmaps for all experimental conditions.** (Full 2D Histogram panels) All pixels in the ground truth test set are plotted against all spatially corresponding pixels in the predicted (test) set according to pixel intensity values and colored by density. A perfect prediction would correspond to a heatmap with all values on the 45-degree line. All plots display raw data without any histogram normalization. Axes are representative of camera bit depth (14- or 16-bit), with some biomarkers only filling a portion of the dynamic range. (Feature Regions panels) Represents a zoomed-in view of the data plotted in the left panel, with the ground-truth axis adjusted to show intensity values above the threshold cutoff value utilized when computing *P* (see *Methods*). This demonstrates the prediction correspondence in the positive-feature region, with low-intensity values (including background noise) excluded-which typically comprise the majority of the test set.
(PDF)

**S4 Fig. Representative accuracy results for a dataset trained on the standard (1-stack) U-Net, compared to a network comprised of two U-Nets stacked back-to-back, with residual connections (2-stack).** Training conditions were otherwise unchanged. Accuracy scores, as reported in terms of the modified *P* (see *Methods* were comparable. *N = 1*0,000 sub-image patches for each test condition.
(PDF)

**S5 Fig. Representative accuracy results for datasets trained on the standard (1-stack) U-Net, with and without data augmentation applied (flip, rotation, zoom) (see Methods).** Training conditions were otherwise unchanged, and results are reported on the complete unmodified test set for each experimental condition. N = 10,000 sub-image patched for each test condition.
(PDF)

**S6 Fig. Representative accuracy results for networks trained using the Mean-Squared Error loss function (MSE) compared to the Pearson's Correlation Coefficient loss function (PCC).** The neural network architecture and training conditions are the same, with the exception of the choice of loss function. Accuracy scores, as reported in terms of the modified $P$ (see *Methods*) were comparable. $N = 10,000$ sub-image patches for each test condition.
(PDF)

**S7 Fig. Cross-platform evaluation.** A network was trained using data collected at 5x on a Zeiss microscope with a 6.5 μm/pixel camera, and used to process new images collected on a Nikon microscope at 4X with a 7.3 μm/pixel camera. We pre-processed data from the new system (Nikon) by scaling it to match the μm/pixel resolution of data from the original system (Zeiss) using the 'Inter-System Correction Factor' to rescale images in ImageJ. Representative ground truth nuclei from MDCK WT cells images on the Nikon system are shown next to the corresponding cross-platform predictions, resulting from both scaled and unscaled input images. Images were contrast adjusted for reproduction by normalizing the histograms and shifting the lower bound of the histogram up by 1/4 of the dynamic range. The scale bar represents 50 μm.
(PDF)

**S8 Fig. Cross-cell-line evaluation.** (A, B) Representative images are shown for either the MDCK cell line or the HUVEC cell line, imaged at 20x magnification. (A', B') For each input image, the ground truth nuclei are shown, (A", A''', B", B''') along with predictions from U-Net models fully trained with either the MDCK or HUVEC nuclei data, (A'''', B'''') with corresponding violin plots shown to display average nuclear size, respectively. (C, D) Additionally, the process is repeated for biomarkers lacking a training set across cell types, such as (C', C") VE-cadherin and F-actin predictions produced from MDCK input, or (D') E-cadherin prediction produced from HUVEC input.
(PDF)

**S9 Fig. Representative image splitting times by image size.** The time in seconds it takes to chop a raw image of equivalent height/width (in pixels) into 256x256 pixel$^2$ sub-images.
(PDF)

**S10 Fig. Representative loss functions from the standard U-Net training process.** Early stopping was enabled, so that if the validation loss did not decrease within a set number of epochs, the training process terminated.
(PDF)

**S1 Table. Accuracy and training statistics for all experimental conditions.**
(DOCX)

**S1 Movie. Fluorescence reconstruction microscopy (FRM) on timelapse data.** A phase-contrast timelapse of MDCK cells, imaged at 5x magnification, is shown at left. The center panel displays nuclear predictions produced by the trained U-Net, given individual frames from the phase-contrast timelapse as input. The overlay of the phase-contrast movie and the nuclear predictions is shown at right. Each panel is 0.31 cm x 0.31 cm, and time between frames is 20 minutes. Video compressed for supplement, but raw data available on request.
(AVI)

**S2 Movie. FRM for high-content screening.** A DIC timelapse movie of MDCK cells, imaged at 20x magnification, is shown at left (top and bottom). The top row displays E-cadherin junctions, while the bottom row displays nuclei. Moving left from right, the second-from-left

images are ground truth (actual) fluorescent images of the junctions/nuclei in green, followed by the FRM predictions in red, and finally the merge of the ground truth and predicted images. Predictions are produced by processing the DIC input on the left through a neural network trained on a dataset of matched DIC and fluorescence image pairs. Panel width is approximately 500 μm, and time between frames is 20 minutes.
(AVI)

**S3 Movie. FRM for fine structures, VE-cadherin.** A DIC timelapse movie of HUVEC cells, imaged at 20x magnification, is shown at left; VE-Cadherin ground truth is shown center; and VE-Cadherin feature predictions are shown at right. Individual panel width is 890 μm, with 20 minutes per movie frame.
(AVI)

**S4 Movie. FRM for fine structures, Nuclei.** A DIC timelapse movie of HUVEC cells, imaged at 20x magnification, is shown at left; nuclei ground truth is shown center; and nuclei feature predictions are shown at right. Individual panel width is 890 μm, with 20 minutes per movie frame.
(AVI)

**S5 Movie. FRM for fine structures, F-actin.** A DIC timelapse movie of HUVEC cells, imaged at 20x magnification, is shown at left; F-actin ground truth is shown center; and F-actin feature predictions are shown at right. Individual panel width is 890 μm, with 20 minutes per movie frame.
(AVI)

**S6 Movie. Mitotic division prediction.** A neural network is able to capture rare events, such as cell divisions, when trained on a sufficiently large and varied dataset. Left panel: a fluorescent timelapse of stained HUVEC nuclei, imaged at 20x magnification. Center panel: the U-Net predictions from DIC images of the same spatial region. Right panel: an overlay of the left and center panels for comparison. Time between frames is 20 minutes.
(AVI)

## Acknowledgments

Special thanks to Gawoon Shim for assistance with HUVEC and keratinocyte data collection.

## Author Contributions

**Conceptualization:** Daniel J. Cohen.

**Data curation:** Julienne LaChance, Daniel J. Cohen.

**Formal analysis:** Julienne LaChance, Daniel J. Cohen.

**Funding acquisition:** Daniel J. Cohen.

**Investigation:** Julienne LaChance, Daniel J. Cohen.

**Methodology:** Daniel J. Cohen.

**Project administration:** Daniel J. Cohen.

**Resources:** Daniel J. Cohen.

**Software:** Julienne LaChance.

**Supervision:** Daniel J. Cohen.

**Validation:** Julienne LaChance.

**Visualization:** Julienne LaChance.

**Writing – original draft:** Julienne LaChance, Daniel J. Cohen.

**Writing – review & editing:** Julienne LaChance, Daniel J. Cohen.

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
