## [Decision Letter · Decision Letter 0]

6 Aug 2020

Dear Prof. Cohen,

Thank you very much for submitting your manuscript "Practical Fluorescence Reconstruction for Large Samples and Low-Magnification Imaging" for consideration at PLOS Computational Biology.

As with all papers reviewed by the journal, your manuscript was reviewed by members of the editorial board and by several independent reviewers. In light of the reviews (below this email), we would like to invite the resubmission of a significantly-revised version that takes into account the reviewers' comments.

We cannot make any decision about publication until we have seen the revised manuscript and your response to the reviewers' comments. Your revised manuscript is also likely to be sent to reviewers for further evaluation.

Sincerely,

Daniel A Beard

Deputy Editor

PLOS Computational Biology

Daniel Beard

Deputy Editor

PLOS Computational Biology

Reviewer's Responses to Questions

**Comments to the Authors:**

Reviewer #1: “Practical Fluorescence Reconstruction Microscopy for Large Samples and Low-Magnification Imaging” seeks to provide guidelines for developing deep learning models that can predict epifluorescence images of labeled biological structures from brightfield images. Unlike prior works, they focus primarily on low magnification (5X-20X single focal frame) images that are common in high content imaging. They demonstrate that despite the lower resolution, U-Net architectures are able to adequately perform this task in a variety of settings, including using reconstructed images for segmentation and tracking.. They show that the most commonly used metric for benchmarking - the pearson correlation coefficient - has several drawbacks that prevent it from being interpreted in a straight-forward manner. Although it is on a scale from 0 to 1, a PCC value of 1 is hardly ever achieved. Moreover, high values of PCC may not correspond to a more accurate reconstruction. The authors present a revised version of PCC, which they call the corrected accuracy score, in which the PCC is computed only on a subset of images that contain positive examples. The authors also investigate the relationship between dataset size and performance, and show that there is little performance gain in using residual U-Net architectures or in using an alternative loss function (e.g. PCC vs MSE). The authors make the models and the underlying source code available. The work presented by this paper fills an existing need in the field, as many of the existing reconstruction methods have led to models geared towards analyzing small amounts of high quality data. Equally valuable, however, is the case of analyzing large amounts of lower resolution data. While the practical advice in this paper is sure to be of value to experimentalists seeking to use these methods, the paper has several issues that should be addressed prior to publication.

Major Concerns

-One of the main focuses of the paper is how to properly benchmark FRM methods. This is important, and the author’s highlight significant shortcoming in the PCC metric. This metric suffers from a classic issue with data centered benchmarking - it is overwhelmed by easy examples. In this case, the authors argue (rightly so) that the “easy” example is the background. A model that can distinguish background from foreground will falsely report a high PCC because the existence background and foreground will effectively induce a correlation (similar to Simpson’s paradox). Benchmarking datasets that have high numbers of background pixels will be prone to this bias. The authors offer an alternative metric - the corrected accuracy score - in which the PCC is computed only on images that have positive examples of the structures in question.

This approach has two issues. First, this appears to be a half measure. I believe a better correction would be to segment foreground from background pixels and then compute the PCC only on the foreground pixels. Choosing images with only positive examples doesn’t resolve the issue, because those images can still contain background pixels. Computing the PCC on the foreground (e.g. cellular) pixels would resolve this issue, as most uses of FRM seek to resolve subcellular structures. Annotation of foreground/background pixels - either manual or computational - should be performed, and the data reanalyzed with these annotations take into account. If this approach is successful, the authors should explicitly define a new metric in their resubmission, to show the connection to and departure from PCC.

The second issue is that FRM does require annotation. The need for annotated training data has been a significant barrier to the adoption of deep learning methods in the life sciences, and the reluctance to embrace this challenge has actually shaped the space of methods developers. While FRM is a great method, the reason why it is popular is because it has a low annotation burden. It only requires the collection of image pairs (label free and fluorescence), and does not need dense pixel level annotations for model training. As this paper reveals, however, annotation of some form is necessary for benchmarking. Without proper, unbiased benchmarks it is impossible to know what is an advance and what is not. Commentary on this point would make the paper considerably stronger, as it would better clarify the weakness of PCC as a metric and why better metrics have not been developed.

-The usability of the software package is less than what I would have expected for a software paper. A Dockerfile should be included, and a pre-built docker image should be posted on Dockerhub. The dependencies are given but are not versioned; this makes setting up a working installation significantly harder. Moreover, the software pipeline as described is (in my view) unwieldy. Key steps for processing data are separated out into different languages (image cropping in fiji, model training/processing and image stitching in python). I feel like this should be done all in one language - a TensorFlow add-on to ImageJ exists, while image cropping is straightforward to handle in python. If the authors choose to go the python route, object oriented programming would be an effective way to organize the different operations that have to be applied to an image (e.g. image resizing, normalization, cropping, model processing, stitching). A GUI would make a Python package more useable (see e.g. NuSet from Jan Liphardt on biorxiv) but wouldn’t be a hard requirement for resubmission. Barring that, a single pre-executed Jupyter notebook accessible on the repository’s README demonstrating analysis with a pre-trained model would be a considerable improvement.

Other issues

-The analysis of model performance vs dataset size is interesting, but it would be more useful if this was couched as # of cells vs # of images. This should be straightforward to compute given that brightfield images can be converted into nuclear images. The comment that this relationship is non-linear is somewhat unnecessary, as non-linearity is implied by the PCC and the corrected accuracy score being bounded above by 1. More important is the monotonicity of the relationship - while there are diminishing returns, better data leads to better models. This should be highlighted in the paper.

-Training for the model performance vs dataset size analysis was performed without data augmentation. While this is understandable for reproducibility, it makes the analysis considerably less informative, as data augmentation can increase the diversity of a given dataset. This analysis should be redone with proper augmentation (rotation, reflection, and zoom) and presented. The non-augmented analysis could be removed or placed in the supplement, as the comparison would allow readers to see the value of augmentation.

-The importance of spatial scale for model performance has been described previously in the context of segmentation (see Van Valen et al PLOS CB 2016 and the recent NuclAIyzer paper from Peter Horvath’s group Cell Systems 2020). One of these works should be cited and this insight should be placed in context.

-Pre-trained models should be made available via TensorflowHub.

-Benchmarking of inference speed (images/second for 1-megapixel images) for the model should be reported. This should also include the time required to crop and stitch images

-Source code should contain a Dockerfile.

-A description of the deep learning software stack (e.g. Tensorflow/Keras vs Pytorch) should be included.

Figure 2 L-M was hard to see - it would be probably best to move this to a supplement so they can be seen at full size

-While the authors claim model architectures with more capacity (e.g. residual U-Net) did not provide a boost in performance, it would be interesting to explore models with reduced capacity. If a mobilenet/FPN was used for feature extraction instead of the conv/max pool from U-Net, would the performance be compromised? Models that are lighter weight but retain accuracy are more valuable, as they are more scalable and can have reasonable throughput if used on CPUs.

Reviewer #2: the review is uploaded as an attachment

**Have all data underlying the figures and results presented in the manuscript been provided?**

Reviewer #1: Yes

Reviewer #2: Yes

PLOS authors have the option to publish the peer review history of their article (what does this mean?). If published, this will include your full peer review and any attached files.

Reviewer #1: No

Reviewer #2: **Yes: **Assaf Zaritsky
---

## [Decision Letter · Decision Letter 1]

16 Oct 2020

Dear Prof. Cohen,

We are pleased to inform you that your manuscript 'Practical Fluorescence Reconstruction for Large Samples and Low-Magnification Imaging' has been provisionally accepted for publication in PLOS Computational Biology.

Best regards,

Daniel A Beard

Deputy Editor

PLOS Computational Biology

Daniel Beard

Deputy Editor

PLOS Computational Biology

Reviewer's Responses to Questions

**Comments to the Authors:**

Reviewer #1: The author's have done a commendable job responding to my comments. I believe the paper is ready for publication.

Reviewer #2: The authors have addressed my concerns and made this elegant and well-written manuscript, even stronger. The paper should be accepted for publication.

Assaf Zaritsky, Ben-Gurion University of the Negev, Israel

**Have all data underlying the figures and results presented in the manuscript been provided?**

Reviewer #1: Yes

Reviewer #2: Yes

PLOS authors have the option to publish the peer review history of their article (what does this mean?). If published, this will include your full peer review and any attached files.

Reviewer #1: No

Reviewer #2: **Yes: **Assaf Zaritsky

---

## [Editor Report · Acceptance letter]

11 Nov 2020

PCOMPBIOL-D-20-01123R1 

Practical Fluorescence Reconstruction for Large Samples and Low-Magnification Imaging

Dear Dr Cohen,

I am pleased to inform you that your manuscript has been formally accepted for publication in PLOS Computational Biology. Your manuscript is now with our production department and you will be notified of the publication date in due course.

With kind regards,

Nicola Davies
